# Detection Based on Crack Key Point and Deep Convolutional Neural Network

**Dejiang Wang \*** , **Jianji Cheng and Honghao Cai**

Department of Civil Engineering, Shanghai University, Shanghai 200444, China; jianjicheng@shu.edu.cn (J.C.); seekercai@shu.edu.cn (H.C.)

**\*** Correspondence: djwang@shu.edu.cn

**Abstract:** Based on the features of cracks, this research proposes the concept of a crack key point as a method for crack characterization and establishes a model of image crack detection based on the reference anchor points method, named KP-CraNet. Based on ResNet, the last three feature layers are repurposed for the specific task of crack key point feature extraction, named a feature filtration network. The accuracy of the model recognition is controllable and can meet both the pixel-level requirements and the efficiency needs of engineering. In order to verify the rationality and applicability of the image crack detection model in this study, we propose a distribution map of distance. The results for factors of a classical evaluation such as accuracy, recall rate, F1 score, and the distribution map of distance show that the method established in this research can improve crack detection quality and has a strong generalization ability. Our model provides a new method of crack detection based on computer vision technology.

**Keywords:** crack detection; deep convolutional neural network; object detection; crack key point; fusion and filtration of features

## 1. Introduction

Cracks are critical flaws that affect the behavior and durability of structures, which can have a negative effect on structural safety. Due to the inevitability and general of cracks on the surface of concrete structures, the search for efficient and low-cost crack detection of concrete has been important in structural damage identification. There are two main directions for the research on crack detection methods: the one is through sensors to test a static and dynamic response of the structure, based on which, the position and depth of a crack are identified [1–3]; the other is through image processing techniques to provide the position and other information about a crack [4,5].

Image-based methods are simple and effective, so they have gained extensive attention. Computer image processing and vision technology, as well as the upgrading of computing hardware and image-based crack detection methods, especially those based on deep convolutional neural networks, have undergone unprecedented development.

Classical image crack detection methods, such as segmentation by a threshold [6], the edge detection algorithm [7,8], and the morphological filtering method [9], not only identify cracks effectively but also assess parameters such as crack length and width. However, their main work is focused on image processing. Crack detection remains a manual process with low efficiency.

To improve the efficiency of detection, researchers have introduced machine learning to deal with crack features and have established a classifier to realize automatic crack detection [10–12]. Crack detection methods of traditional machine learning algorithms combined with image processing techniques have been applied in this area.

Machine learning has broadened the idea of applying computer vision methods for defect detection and condition assessment in civil engineering [13] and has brought about new research directions for all types of detection, including crack detection. Many

researchers, using these latest machine learning algorithms, have continued to propose novel image crack recognition models [14–17].

Relying on manual extraction of the characteristics of cracks to realize the crack detection of an image cannot meet the needs of a project due to the complex information contained in the actual crack images. In image- and video-based ML approaches for structural health monitoring, differences in illumination, rotation, and the angle of the camera can significantly affect the final results [18,19]. To meet the requirements for crack detection in practical engineering, automatic learning algorithms based on crack features, especially deep convolutional neural network algorithms, have become a research hotspot. These methods eliminate the first image processing step of most traditional methods, and, based on original crack images, can directly extract crack features and detect cracks through automatic learning models.

Hinton [20] first proposed the concept of deep learning, which has gained extensive attention in the machine learning area. Models based on deep learning began to emerge [21–23]. Using the same dataset, Dorafshan [24] compared the concrete crack detection results of classic edge detection with a deep convolutional neural network (DCNN). The results showed that DCNN had advantages in terms of accuracy, detection speed, and resolution.

Based on the image segmentation algorithm, many detection methods have been proposed [25–28]. Those methods with high accuracy obtain a good detection effect, especially for crack width. However, the above methods are all based on image segmentation algorithms, which require a huge amount of work of pixel-level marking on pictures yet still do not reach the expected accuracy in some cases. A large number of diverse crack training samples are also usually required to achieve better detection results [29].

In recent years, object detection methods based on points have been emerging. Zhou [30] modeled an object as a single point. Hei [31] raised corner detection, while Duan [32] established a method through center pooling and cascade corner pooling, three of which have inspired the research on crack detection method based on key points. Although the method Lee [33] established is still based on pixels, the detection result is more targeted at predicting crack areas rather than pixels. This method has been an inspiration for further research on crack detection based on crack key points that is effective and suitable for engineering.

Crack detection methods based on deep learning depend on the extensity of the training set and the validity of their algorithm. In terms of obtaining crack image data, the current information era ensures easy access to a huge number of surface crack images, so the crack detection methods based on deep learning are reasonably trustworthy. There are two main directions for crack detection algorithms. One regards crack detection as an object segmentation task by image segmentation algorithms to classify and predict pixels and finally output binary images [34]. This method can precisely predict the location and width of cracks but requires detailed and accurate annotation of crack images. However, crack detection in practical engineering works does not require pixel-level positioning. Another strategy is to simplify the problem. This method partitions the crack image and detects crack individually on each patch before stitching them together to locate the cracks [22]. This method simplifies the annotation and can use object sorting algorithms with high convergence. Nonetheless, this method causes two problems: the lack of integrity of cracks when partitioning and the loss of the global feature of the crack image, reducing the generalizability and noise immunity of the model.

Compared with the usual visual computer tasks, crack detection poses three notable challenges. Firstly, cracks are typical linear objects. Secondly, the commonly existing environmental noise causes confusion, so the crack detection results are sensitive to global features. This crack detection problem is relatively severe, influencing the detection results and invalidating the traditional image processing algorithms. Third, the task is substantially a binary classification problem with only two options, having a crack or not. Therefore, only these specificities are considered when designing and modifying a conventional deep convolutional neural network model. The accuracy and suitability of the crack detection

method are therefore called into question. Some studies have also shown that the crack feature information of different scales greatly influences the crack identification effect.

For this purpose, we established a crack detection model that considers the cracks' linear characteristics and maximizes the use of their global characteristics. Based on the cracks' linear characteristics, this method can identify cracks as long as the key points on the cracks are identified. Considering the network advantages of traditional computer vision tasks, this research work uses the separation and fusion of the global and local features of cracks to construct a KP-CraNet model for crack detection. Finally, evaluation criteria are set to evaluate the effectiveness and suitability of the model. The numerical experiment has proven that our crack detection model, KP-CraNet, showed a relatively strong detection ability with great potential for further improvement.

## 2. Crack Characterization

### 2.1. Crack Key Point

Cracks are gaps with a certain width presented as melanic pixel point sets in a crack image in physical space. We try to realize the second-level evaluation task proposed in [35]—that is, to find the expression of the geometric position of crack damage. Most of the current image-based crack detection methods predict cracks based on pixel points and use the binary image to present cracks, such as the surface crack [36] shown in Figure 1a, and its detection result binary image is Figure 1b. The crack binary image detection takes full advantage of the image pixel information and precisely expresses the location, length, width, and other information about a crack. However, most crack detection works do not require pixel-level information about a crack in reality. Additionally, for deep learning based on image pixels, the supervised training of the model requires a large number of highly accurate cracks that are manually marked on the image to create a pixel-based crack dataset. The amount of work required to mark every pixel of the image is enormous. The representation of cracks based on crack key points, as shown in Figure 1c, requires the ligature of the adjacent key points to represent the crack location information. Crack key points do not have to be in the crack but can be just near it with a distance required by the engineering detection accuracy. Therefore, there can be more than one crack key point set. For example, the crack is shown in Figure 2a can define two types of key points set in Figure 2b,c. The non-uniqueness of crack key points set may lead to some problems, which will be discussed later in the article.

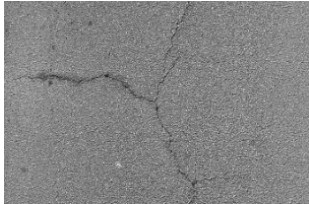 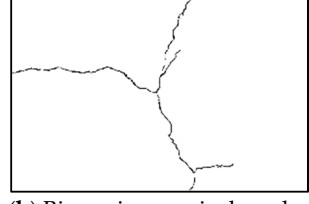 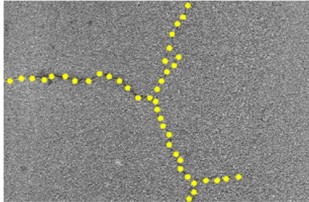

(**a**) Surface crack original image   (**b**) Binary image pixel marked   (**c**) Crack key point marked

**Figure 1.** Two characterizations of cracks.

For a given image, the crack key point set can be defined through Equation (1) based on the crack features:

$$P = \{p_1, p_2, p_3, \cdots, p_N\}, \tag{1}$$

where $N$ is the number of key points; any element in the set is a subset as given by Equation (2):

$$p_i = \left\{ s_i^{\text{front}}, s_i^{\text{back}}, (px_i, py_i) \right\} \tag{2}$$

The subscript $i$ is the sequence number of a crack key point; $s_i^{\text{front}}$ is the sequence number of its preceding key point, and $s_i^{\text{back}}$ is the latter; and $(px_i, py_i)$ is the pixel coordinate of the crack key point. The sequence numbers of the preceding and latter key points

demonstrate the connection type of crack key points, the ligature of which provides the location information of the crack.

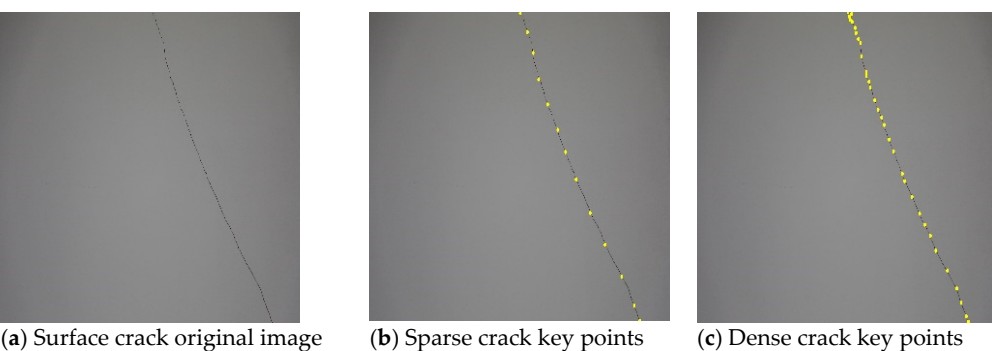

(**a**) Surface crack original image     (**b**) Sparse crack key points     (**c**) Dense crack key points

**Figure 2.** Different crack key points can represent the same crack.

### 2.2. Reference Anchor Point Method

The non-uniqueness of crack key points means three problems need to be solved. Firstly, the crack detection model clarifies the relevance of detected key points to predict cracks correctly, regardless of their non-uniqueness. Secondly, whether sparse or dense, the key points are distributed in the crack area; thirdly, we have to control the number of detected crack key points, because it prejudices the model training when there are too many or too few.

#### 2.2.1. Set-Distance Scattering

When detecting cracks, the actual pixel distance of crack key points is hard to set to a universal standard. For cracks with high sinuosity, their key points should be dense, while for cracks with low sinuosity, such as linear types, several key points should be enough.

To avoid the crack detection results being influenced by the sparse key points, it is important to scatter the key points evenly to keep the distances between adjacent key points the same. If the distance between any two adjacent key points is smaller than a certain threshold, no operation is needed; if it is longer, new key points need to be found on their ligature at the same distance as the others to be part of the original crack key point set, forming a new one, to make sure the distance between any two adjacent key points is smaller than a threshold. This process is shown in Figure 3.

For adjacent key points, when their ligature is shorter than a pre-set threshold, no operation needs to be done; when it is longer than the pre-set threshold, then the ligature of the two points needs to be scattered to insert points at equal distance, as shown as Equation (3):

$$M = \left\lceil \frac{\text{dis}(p_i, p_j)}{\varepsilon_{\text{s}}} \right\rceil, \tag{3}$$

where $\text{dis}(p_i, p_j)$ is the pixel distance between adjacent crack key points $p_i$ and $p_j$, while symbol $\lceil \rceil$ means rounding up to an integer.

We detected the $M - 1$ inserted points to determine whether they were crack key points, and traverse preceding and subsequent key point pairs to form a new crack key point set, as in Equation (4):

$$P^{\text{label}} = \{(x_1, y_1), (x_2, y_2), \cdots, (x_{N^{\text{label}}}, y_{N^{\text{label}}})\}, \tag{4}$$

where $N^{\text{label}}$ is the number of crack key points after set-number scattering in the current image, while $P^{\text{label}}$ is the crack key point set in a single image of the model we proposed.

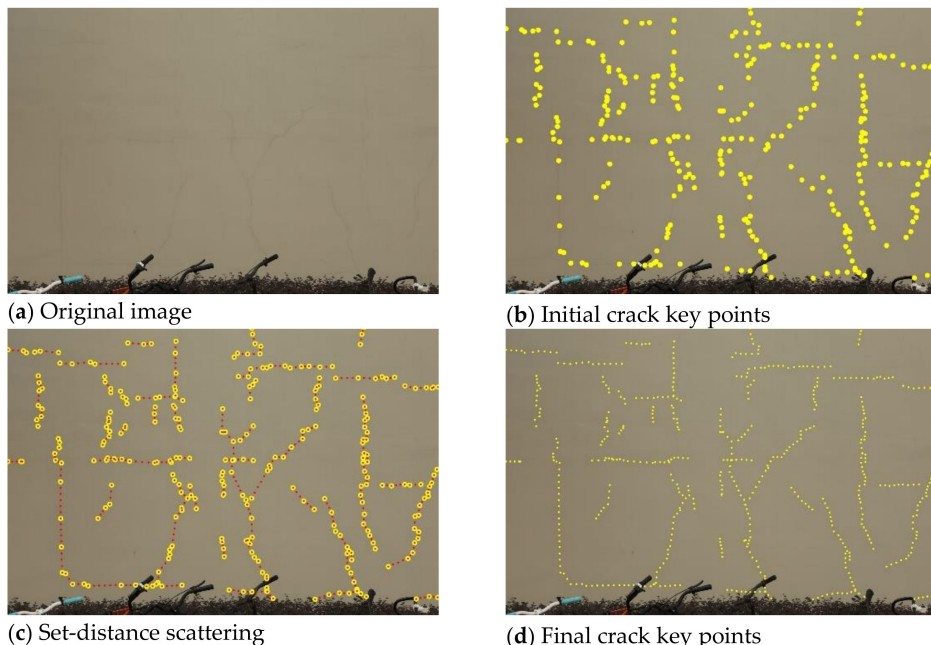

(**a**) Original image

(**b**) Initial crack key points

(**c**) Set-distance scattering

(**d**) Final crack key points

**Figure 3.** The process and result of set-distance scattering.

### 2.2.2. Set Reference Anchor Point

Although the sparsity of crack key points can be controlled through set-distance scattering, which reduces the error caused by the density of crack key points to an extent, the number of crack key points is still a problem in real-life detection. If the number of crack key points is too low, it may result in an inability to identify all cracks accurately; if the number is too large, it is very computationally intensive. Therefore, this research proposes a reference anchor point method for crack identification.

The reference anchor point method is derived from the anchor mechanism of the prediction box in the faster R-CNN model [37]; in this research, the anchors were considered the prediction points for crack detection. We laid out the anchor points on the image in advance and detected whether each anchor point was near a crack key point to keep only the nearest anchor points as the final detection results. This method can effectively solve the latter two problems mentioned above.

We set the reference anchor point as in Equation (5):

$$P^{\text{anchor}} = \{(x_1, y_1), (x_1, y_1), \cdots, (x_{N^{\text{anchor}}}, y_{N^{\text{anchor}}})\}, \tag{5}$$

where $N^{\text{anchor}}$ is the number of the anchor point of a single image, and $N^{\text{anchor}}$ is related to the anchor point layout. The distance between the adjacent two anchor points $S$ is the only parameter for determining the anchor point location, i.e., the anchor point stride. The reference anchor point method is used to set anchor points in an image based on a certain pixel distance in advance, as the red dots show in Figure 4. Thus, crack detection has turned into a matter of calculating the probability value of reference anchor points as crack key points, and when the probability value is higher than a certain threshold, the anchor point will be regarded as a detected crack key point.

The number of anchor points varies according to the actual situation, enabling the reference anchor point method to accurately detect cracks based on anchor points in different densities. For example, the crack area is first determined using anchor points with large step lengths, and then the local crack area obtained is refined to make a more accurate prediction of the cracks using dense anchor points, thus establishing a multiscale crack identification method.

The interval between the anchor step and the crack key determines the crack identification accuracy. If the anchor step is in pixel units, this crack-critical point method

is almost equivalent to the pixel-based image crack identification method (conventional image segmentation method). They differ only in terms of the labeling error in crack labeling and whether the pixel width of the crack is considered. On the other hand, if the entire cracked image has only one crack key, the method is equivalent to the image classification task, which divides the whole image into areas with or without cracks. Thus, the image segmentation and image classification methods are special cases of the crack key point method.

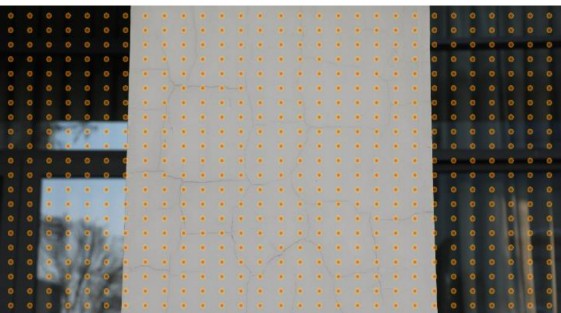

**Figure 4.** Reference anchor layout.

### 2.3. The Determination of Positive and Negative Sample Point

During the model training for crack detection, the crack key points need to be marked in the training set images as marking points. The training process involves the model learning to detect whether a reference anchor point is a crack key point (also called a marked point). Not all crack key points coincide with reference anchor points, so reference anchor points can be classified into three types: positive sample points, negative sample points, and general anchor points. A positive sample point is defined as the closest anchor point to the crack key points. For each crack key point $p_j^{\text{label}}(j = 1, 2, 3, \cdots)$, its positive sample point is shown in Equation (6).

$$p_{ji}^{\text{anchor}} = \min_i \left( \text{dis}\left( s_{ij} \right) \right), \tag{6}$$

where $\text{dis}\left( s_{ij} \right)$ is the distance between the anchor point $p_i^{\text{anchor}}$ and the crack key point $p_j^{\text{label}}$.

However, to maintain the convergence of the model training, all anchor points whose distance to a crack key point is shorter than a certain threshold, $\varepsilon_{\text{P}}$, will be set as a positive sample point. So, the positive sample point is set as in Equation (7):

$$E^{\text{T}} = \left\{ p_i^{\text{anchor}} \middle| \text{dis}_{ij} < \varepsilon_{\text{P}}, i = 1, 2, \cdots N^{\text{anchor}}; j = 1, 2, \cdots N^{\text{label}} \right\} \tag{7}$$

So, the reference anchor points in an image belong to a positive sample point set, a negative sample point set, and a general anchor point set. Figure 5 shows the crack key points (green), positive sample point set (red), and negative sample point set (blue) when $\varepsilon_{\text{N}} = \varepsilon_{\text{P}}$.

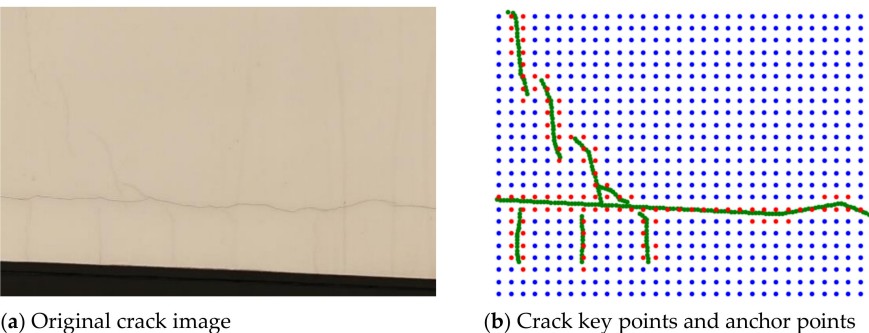

(**a**) Original crack image          (**b**) Crack key points and anchor points

**Figure 5.** Positive and negative sample points.

## 3. Crack Detection Model KP-CraNet

To consider the influence on crack detection exerted by the linear and global features of cracks, a crack detection model based on crack key point, KP-CraNet, has been built. The model contains three hierarchical submodels. The bottom submodel is a feature extraction model based on crack key points of deep convolutional neural networks, which is also called a key point feature extraction network or basic network. The second submodel uses feature pyramid fusion and a reinforce network [38] (FPN feature fusion and reinforce network) to fuse and reinforce the features of extracted key points. The third submodel is a feature filtration network, which filters the features of crack key points, the results of which will lead to an area of a crack key point as an approximate location of a crack.

### 3.1. The Network Frame of KP-CraNet

Figure 6 shows the network frame of KP-CraNet. The bottom submodel for crack key point feature extraction adopted the ResNet [39] network framework, choosing the last three feature layers with different sizes, R1, R2, and R3, as the feature fusion input and reinforce networks. In this network, the FPN reinforcement network is used for global and local feature fusion. The output results are dominated by the current layer features, and the upper layer feature layer is incorporated into the current layer using inverse convolution, outputting a total of five layers. The first three layers have the same size as R1, R2, and R3, while the last two layers are based on the R3 layer, again convolved and downsampled to obtain a smaller layer to better express the global features. Finally, the five feature layers are subjected to feature filtering. The features of feature layers 1, 3, and 5, which are in decreasing size order, are inputted into the feature screening submodel, and the probability of predicting the anchor point as a crack key point is used to screen the positive sample points sequentially to detect cracked and noncracked areas for the final detection of cracks.

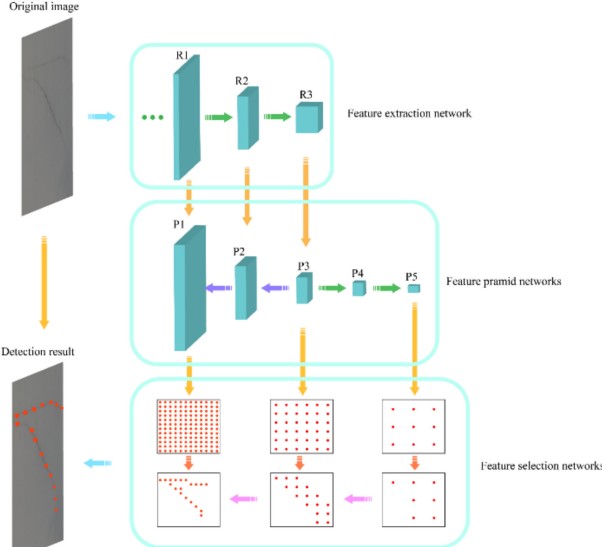

**Figure 6.** Model network structure.

### 3.2. Crack Key Point Feature Extraction Network

The key point feature extraction network uses deep convolutional neural networks, usually adopting ResNet or VGG or other models commonly used in the computer vision field. Crack detection experiments show few differences between the ResNet and VGG models, even when increasing the network depth. Therefore, ResNet18, which requires lower depth, was chosen to decrease the training difficulty and improve the prediction efficiency.

The input to the lowest layer is the image, and the information contained in each pixel of it is the smallest unit of the feature, called the smallest local feature. The topmost layer feature is the crack identification result, and all intermediate result layers from

the bottommost to the topmost are called feature layers. For any current layer, whether it is convolution or pooling, the feature points within the range of $3 \times 3$ convolution kernels or $2 \times 2$ pooling kernels of the current layer are weighed and summed, and the activation function is used to obtain new information about the feature points. Thus, each feature point of the next layer contains multiple feature point information of the corresponding position of the current layer. The feature information is continuously downscaled and integrated, gradually transitioning from local feature information to global feature information.

Figure 7a,b shows the convolution and pooling layers. The rear, lighter part is the current layer feature map. The front, darker part is the next layer. The images demonstrate how framed feature points in the current layer descend to the next layer, which means the next layer contains information on points in the current layer, so the local feature information is gradually integrated with the layers, adding up. As the feature layer increases, the area of information contained in feature points becomes larger. The red dots in Figure 8 are feature points, and the yellow-green background is the feature area. In a higher-level feature map, the feature points reflect global features.

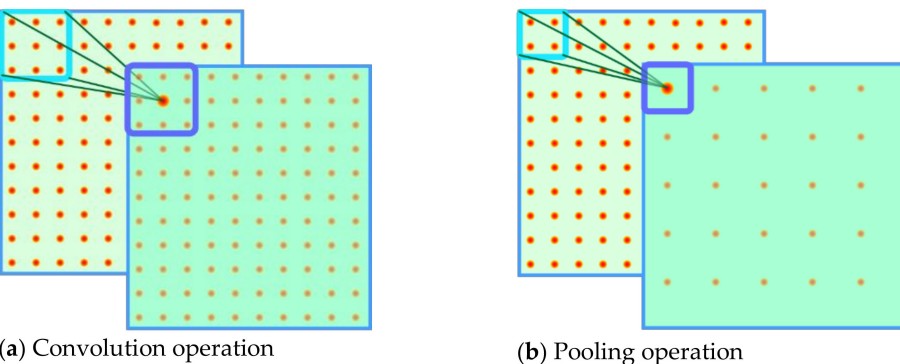

(**a**) Convolution operation　　　　　　　　　　　(**b**) Pooling operation

**Figure 7.** The extraction process of convolution and pooling layers.

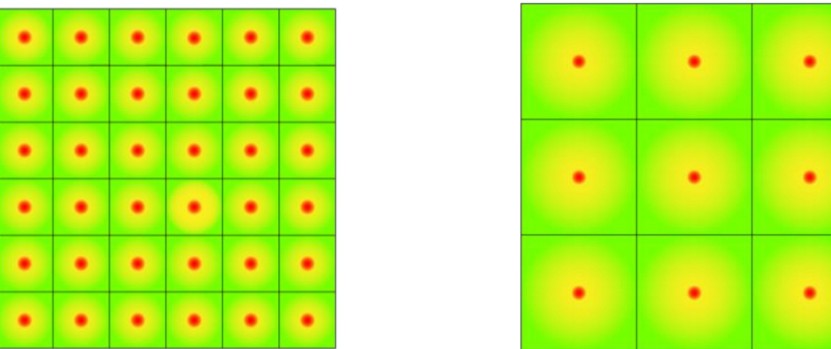

(**a**) Feature points and area of current layer　　　　(**b**) Feature points and area of next layer

**Figure 8.** Feature points and feature area.

### 3.3. FPN Feature Fusion and Reinforce Network

Three maps of features from the feature extraction network contain both global and local feature information. Although the specific location of cracks should be precisely positioned by a local feature map, we also need global features as a reference to avoid the errors caused by tree branch shadows and stains that look similar to cracks in pictures. Although, in the feature extraction of single-track networks, higher feature layers contain some of the features of lower layers, the same layers may contain the feature information of different layers in different images because of the varying crack rate. Therefore, it is important to consider both global and local features. Feature information in different

layers should be further fused and reinforced to obtain integrated feature information that contains features through all layers. FPN feature fusion and reinforce network is an effective method of feature fusion and reinforces various sizes that are extensively applied in the object detection field. After the convolution and fusion of the three feature maps extracted by the network, three new feature layers are generated, with convolution and pooling operations repeated twice to form a higher global feature layer, so the FPN feature fusion and reinforce network would generate five feature layers in total. Apart from containing more information, it can also process images of different sizes, learn the feature rules automatically, and finally, generate three reinforced feature maps and two with global feature information. We selected the first, third, and fifth layers in sequence as the final extracted feature layer with detailed local features, transition features, and global features.

### 3.4. Feature Filtration Network

The outputs of FPN feature fusion and reinforce network need another convolution operation for feature processing and finally output an anchor point prediction result the same size as the corresponding original feature layers. Different layers contain different numbers of local or global features, so the prediction results of each layer can be regarded as the prediction results for different features' extent, i.e., whether the corresponding area contains a crack.

Predictions that consider global features will have higher prediction accuracy, while predictions considering local features will have higher localization accuracy. Thus, starting with global features, the feature points that do not have cracks are gradually filtered out based on the results of the current layer prediction. Finally, the identification results are obtained, and the prediction accuracy and crack location accuracy are guaranteed. The process of crack detection shown in Figure 9 exemplifies how a feature filtration network works.

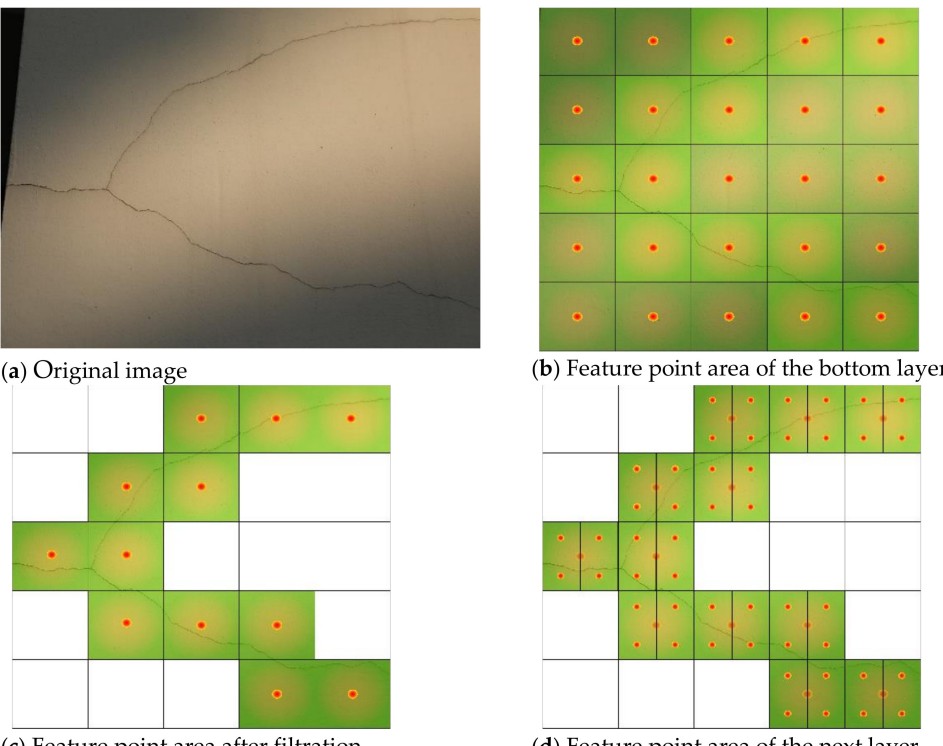

(**a**) Original image          (**b**) Feature point area of the bottom layer

(**c**) Feature point area after filtration          (**d**) Feature point area of the next layer

**Figure 9.** Feature filtration network.

Firstly, we input the prediction results of the bottom feature layer that contains the most comprehensive global information, as shown in Figure 9b, into the feature filtration network. Due to the relatively large distance between feature points, each feature point

contains comprehensive global feature information. The prediction result of the current layer helps to distinguish between the crack area and noncrack area, as shown in Figure 9b. We removed the feature points in noncrack areas and kept those in the crack area, as shown in Figure 9c.

Based on this filtering result, the feature points corresponding to the next feature layer were retained. As shown in Figure 9d, the features and anchor points of this layer were screened, and after the screening, only the anchor points of the current layer in the crack-containing area were retained again; this process was repeated continuously. According to the network structure, three times in sequence were filtered, and the points finally retained were the anchor points of the positive sample of the lowest local features at the maximum resolution set and the minimum step size. These were the key points of the identified cracks.

It is important to note that the above method is completely different from directly deciding whether an area contains cracks. The method depends totally on convolution or pooling operations, whose weight is obtained via the model's automatic learning.

### 3.5. Feature Filtration Network

In model training, the input for each layer level is the anchor point prediction result of the previous layer of features and the eigenvalue of the current layer of features. The output is the anchor point prediction result of the current layer, which is used to determine whether each anchor point is a positive sample point of the current layer. Due to the different sizes of the input images at different layers, or the different sizes of the feature layers, the tolerance of the critical point determination for identifying cracks is also different. So, the thresholds $\varepsilon_P$, $\varepsilon_N$, and loss function are determined by the size of feature maps.

For reticulated reference anchor points at equal intervals, if their anchor point stride is S pixels, then the maximum distance between a crack key point and its nearest anchor point is $\sqrt{2}S/2$ pixels. As shown in Figure 10, the yellow and green points are fracture key points, and the furthest possible location from the nearest anchor point to that fracture key point is the four red points around it. $\varepsilon_P = 0.8S$ and $\varepsilon_N = 1.5S$ can be extracted to define positive and negative sample anchor points in every layer. When the possibility that the reference anchor point is a positive sample point is higher than the set threshold, the reference anchor points are regarded as positive sample points of the current layer.

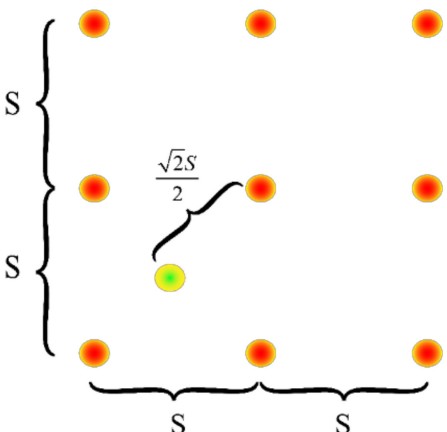

**Figure 10.** Set the distant threshold of positive and negative sample points.

A loss function is introduced in each layer to speed up the algorithm convergence and improve the judgment accuracy during training. For this classification problem, a binary cross-entropy loss function can generally be used. However, for image crack detection problems, because a crack occupies only a small part of the whole image, there are many more negative sample points than positive ones, which will lead to error. Therefore, a loss

function in RetinaNet [40] is applied to balance the positive and negative sample points, using the Focal Loss function as in Equation (8):

$$f_{\text{loss}} = \begin{cases} -\alpha(1-y)^\gamma \log y, & \text{when } \widetilde{y} = 1 \\ -(1-\alpha)y^\gamma \log(1-y), & \text{when } \widetilde{y} = 0 \end{cases} \tag{8}$$

In order to balance positive and negative samples, regulatory factors α and β were introduced into the formula. $y$ is the predicted value, i.e., the possibility that reference anchor points are crack key points of the feature layer. $\widetilde{y}$ is the actual value of the same reference anchor points: the positive sample point is 1, while the negative one is 0.

## 4. Model Training and Evaluation

### 4.1. Data Collection

The dataset used in this research was derived from the surface crack images of various wall surfaces on a university campus in Shanghai. First, we used a camera to take multiple $6720 \times 4480$ pixel images, divided them into $6 \times 6$ blocks, and used bilinear interpolation to unify each block image to a size of $1024 \times 768$. From these images, 1349 images containing cracks were selected; 90% of them were used as the training set and 10% as the test set (https://github.com/csga11/craData, accessed on 24 November 2021).

### 4.2. Training Parameters

The crack detection model in this paper was based on the Pytorch 1.0 deep learning framework. The GPU used for training and testing was NVIDIA GTX1080. The initial parameter weights of the feature extraction network all used the weights of the Pytorch official ResNet network pretraining model. During training, the batch size was 16, the number of iterations was 100, and the learning rate was 0.0001. In order to improve the training effect and robustness of the model, conventional online dataset expansion methods, such as random inversion, filtering, and brightness enhancement, were used during the model training.

### 4.3. Assessment Criteria

The model assessment criteria need to be determined to evaluate whether a crack detection model is effective. For each anchor point, the distance threshold method was applied to determine whether a sample point was positive or negative. As the possibility of positive sample points was calculated as the output, a possibility threshold $\varepsilon_b$ had to be set for determining the sample points, i.e., the positive and negative sample point set were as in Equation (9):

$$E^{\text{P}} = \left\{ p_i^{\text{anchor}} | y_i > \varepsilon_b \right\}, E^{\text{N}} = \left\{ p_i^{\text{anchor}} | y_i < \varepsilon_b \right\}, \tag{9}$$

where $y_i$ is the predicted possibility value of the anchor point i.

The commonly used accuracy rate in the object detection area was also adopted to represent the correct prediction rate among all the image detection results as in Equation (10):

$$P = \frac{num\{p_i | p_i \in E^{\text{T}}, p_i \in E^{\text{P}}\}}{num\{E^{\text{P}}\}} \tag{10}$$

In Equation (11), $num\{\}$ represents the number of elements in the set.

Meanwhile, the introduction of another conventional indicator-recall rate, the proportion of all actual crack points that are correctly identified, describes whether all the key points of the crack are correctly identified. For any key point of a crack, as long as there is an anchor point identified as a positive sample point in the circle whose radius is step S,

the key point of the crack is considered to be correctly detected. Therefore, the recall rate was defined as in Equation (11):

$$R = \frac{num\{p_j | p_i \in P^{anchor}, p_j \in P^{label}, dis(p_i, p_j) < S\}}{num\{P^{label}\}} \tag{11}$$

It can be seen that, in order to ensure accuracy, forecasts should be as few and precise as possible. However, in order to ensure the recall rate, the prediction should be as complete as possible, so the two indicators are contradictory to a certain extent. According to actual engineering needs, these two parameters can usually be adjusted. Here, the weighted balance F1 score of the two is adopted as the final identification evaluation index. The recall rate is defined as in Equation (12):

$$F1 - score = \frac{2PR}{P + R} \tag{12}$$

This research proposes the concept of a distance distribution map to reveal the distance distribution of the identification point from the nearest crack key point. The abscissa is the distance between the anchor point and the key point of the nearest crack, and the ordinate is the number of anchor points at the above distance. The results of all the images in the test set were plotted in one graph. The quality of crack detection was evaluated by analyzing the number of anchor points within a certain distance.

### 4.4. Analysis of Detection

The correctness of the model design needs to be tested first to demonstrate the effectiveness and applicability of the method. We tested three models without a feature filtration subnetwork, then output three layers, P5, P4, and P3. P5 represents the highest layer for global features. After testing the two characteristic screening subnetwork models, P5 is the screening output of P4, and P5 and P4 are the screening output of P3. We set the threshold rate using the experimental results of the above five models, as shown in Table 1.

**Table 1.** The assessment result of each output layer when the threshold rate set as 0.5.

| Output Layer | Distance Threshold | Accuracy Rate | Recall Rate | F1 Score |
|---|---|---|---|---|
| P5 | 128 | 0.869 | 0.992 | 0.895 |
| P4 | 64 | 0.814 | 0.894 | 0.853 |
| P3 | 32 | 0.786 | 0.882 | 0.831 |
| P5 + P4 | 64 | 0.819 | 0.909 | 0.862 |
| P5 + P4 + P3 | 32 | 0.881 | 0.833 | 0.856 |

When the output layer is a single feature layer, and the sizes of feature layers increase (from P5 to P3), the correctness of prediction gradually drops. The F1 score decreased from 0.895 to 0.831, which showed that the global feature exerted a notable influence on crack detection. So, the global feature should be adequately considered.

However, after adding the feature screening subnetwork, comparing the results of P4 with P5 + P4 and P3 with P5 + P4 + P3 shows that the model has better detection results after adding the feature screening subnetwork. The F1 score increased from 0.853 to 0.862 and from 0.831 to 0.856.

Figure 11 presents a distance distribution map of the detection results of the single-track output layer model. The intensive blue strips are a histogram of the distance between anchor points and the nearest crack key point. This shows that, as the distance between anchor points and the nearest crack key point increases, the number of anchor points reduces. For the orange anchor point segmenting vertical lines in the image, its vertical coordinates are the threshold of positive/negative sample determination for the current feature layer: positive to the left, negative to the right. The curves in different colors are anchor point segmenting lines under different possibility thresholds, and any line divides

all the anchor points (blue strips) within the distance into two parts: the upper parts represent the positive prediction points, and the lower represent the negative prediction points. Any point in the curve represents the number of positive points in the corresponding distance, which can be called a positive detection curve.

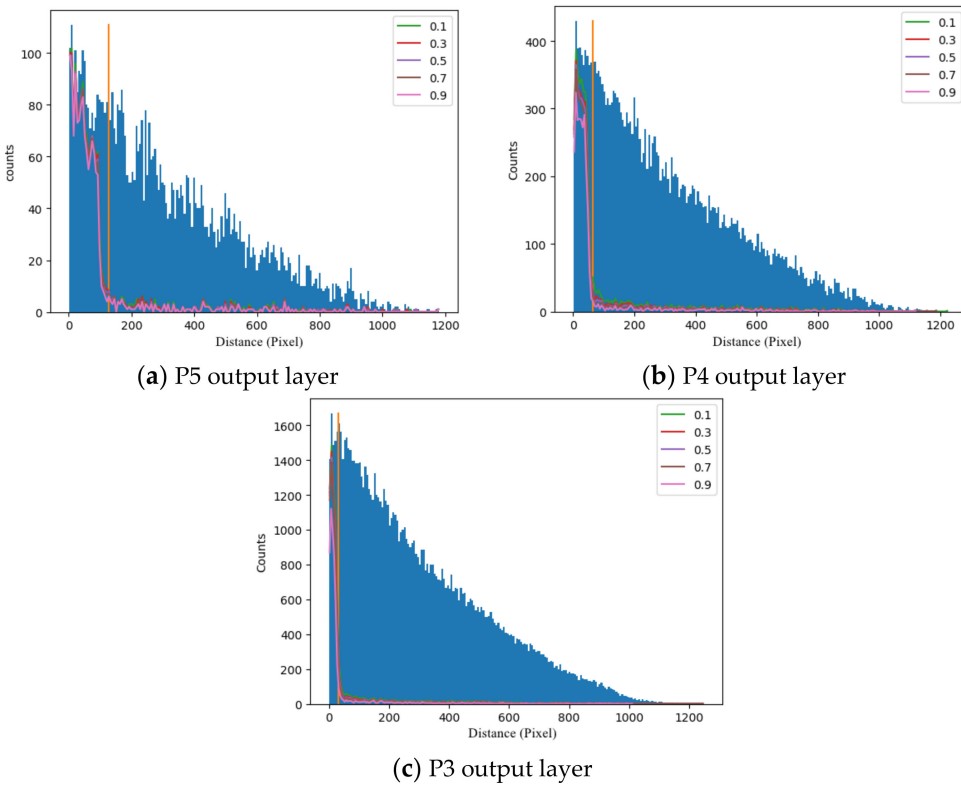

(**a**) P5 output layer    (**b**) P4 output layer

(**c**) P3 output layer

**Figure 11.** The histogram of the shortest distance between anchor points and marked points of single-track output layer.

Any one of the identifying positive case curves and vertical dividing lines divides the blue area into four regions. In the upper left region, the points that are actually positive case anchors are identified as negative case anchors; in the lower left region, the points that are actually positive case anchors are identified as positive case anchors; in the upper right region, the points that are actually negative case anchors are identified as negative case anchors; and in the lower right region, the points that are actually negative case anchors are identified as positive case anchors. The percentage of points in the lower left and upper right areas should be as high as possible.

Therefore, for the P5 output layer shown in Figure 11a, the positive prediction curves under each possibility threshold almost coincide; the threshold selection has little influence on the result, which means that the global feature exerts a relatively significant influence on model detection and the detection of the crack key point is accurate. The prediction rate of anchor points marked as positive is close to 1, while the prediction rate of anchor points marked as negative is close to 0, which is an ideal prediction result. Figure 11b,c demonstrate that, with the decrease in stride, the detection result of the positive/negative sample point becomes more sensitive to the possibility threshold, and more detection errors appear, because images with high resolution emphasize the local features, leading to the misjudgment of anchor points in areas such as cracks. This illustrates that global features are more helpful for effective crack detection, yet the relatively long stride complicated the accurate location of cracks. The introduction of a filtration subnetwork requires a correction of the distance distribution map, because the number of anchor points is influenced by the feature filtration subnetwork and the size of its output. Changing the vertical ordinate from the number of anchor points to the anchor point frequency would reduce the influence of the model anchor point to an extent.

Figure 12 presents the distance distribution map of a single-track output layer and introduces the feature filtration layer under a different possibility threshold. It shows that the curves become steep near the possibility threshold after introducing the feature filtration network, especially the output of P3; more points are detected near the crack key points, and fewer are detected away from the crack key points. The figure illustrates that the feature filtration network has a significant effect, verifying the suitability of the networks we presented.

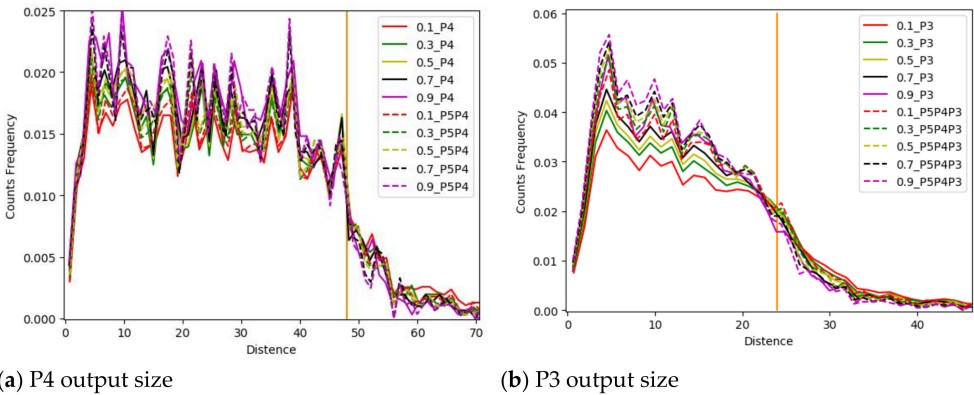

(**a**) P4 output size  (**b**) P3 output size

**Figure 12.** Frequency histogram of distance between single output layer and filtration layer anchor point.

Finally, we chose P5 and P3 as feature filtration layers, while P1 was an output layer. The final results of different crack detection were as shown in Figure 13. This figure proves that the application of our detection model can lead to satisfactory crack detection results. Moreover, the detection experiments discovered that the model had good generalizability, enabling the model to detect manual marking errors. Figure 14a was used to identify cracks. Figure 14a contains much interference, such as cracks, shadows, and numerical values. The model detection result is shown in Figure 14b. The comparison of the two figures shows that the model can detect cracks in a complex environment and meets the basic requirements of crack identification in a practical engineering environment. The suitability of the model design proposed in this research has been verified again. It is worth mentioning that the method proposed in this paper utilizes points but not crack characteristics in the marking stage. Therefore, width recognition of cracks is not accurate enough. We should consider modifying and optimizing this method in future research.

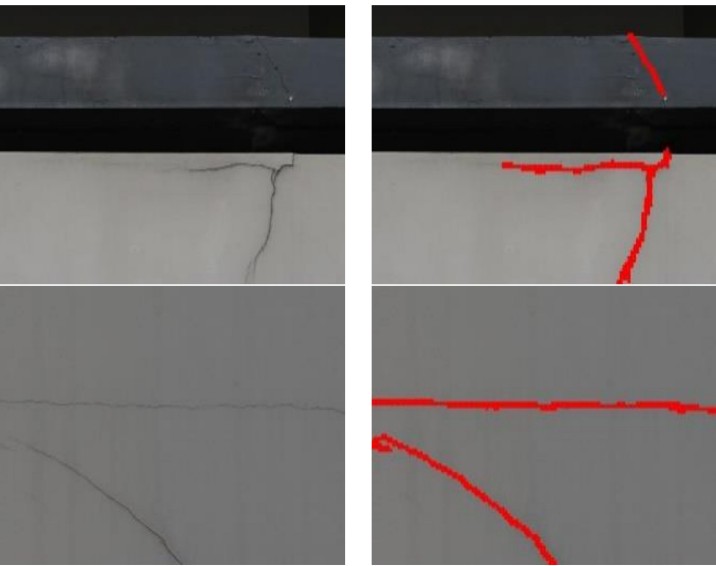

**Figure 13.** *Cont.*

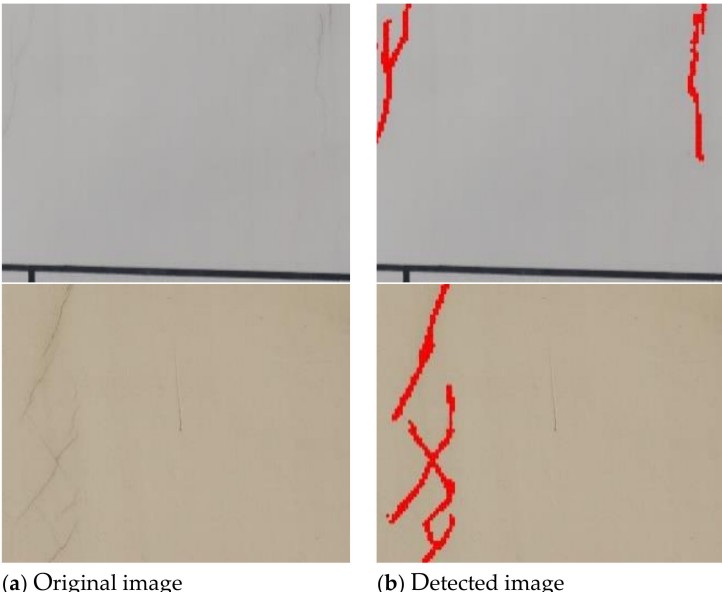

(**a**) Original image        (**b**) Detected image

**Figure 13.** Detection results of different cracks.

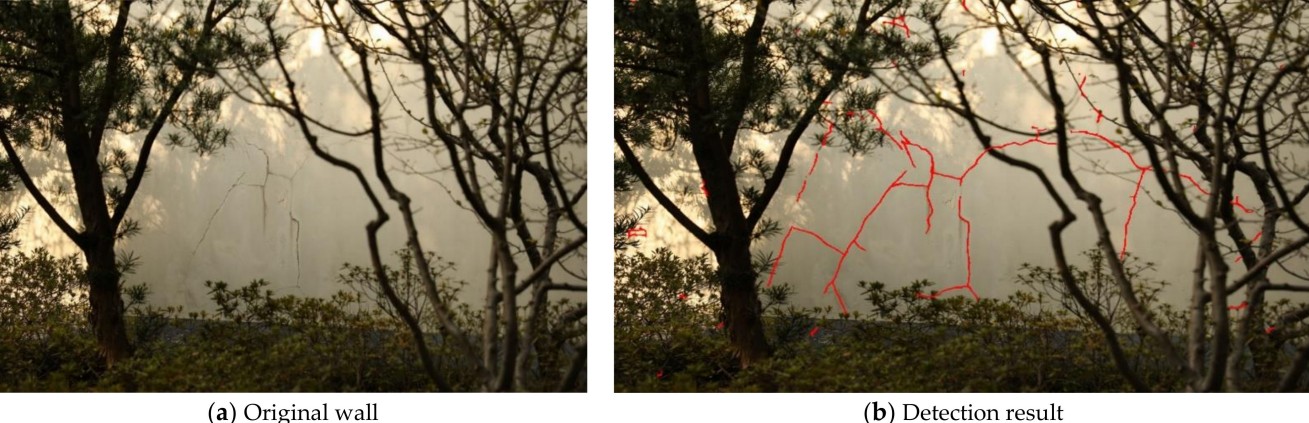

(**a**) Original wall                    (**b**) Detection result

**Figure 14.** Detection of cracks on wall with interference factors.

### 5. Conclusions

Based on the characteristics of cracks, this paper defines the concept of crack key points, combined with the anchor mechanism in computer vision technology, and proposes a new crack identification method—the reference anchor point method. This research established a new model of image crack detection based on deep learning. Through the analysis of the detection network model and the crack detection experimental results, the following conclusions were obtained:

- This research proposed a crack characterization method, combined with the features of image cracks based on key points of cracks. Its detection accuracy is controllable, which can lead to pixel-level recognition effects and can greatly improve detection efficiency based on meeting the accuracy requirements of engineering. When the computer is configured with NVIDIA GeForce GTX 1080, the recognition time of a single photo is 30 ms.
- The advantages of characterizing image cracks based on the key points of cracks are explained. By designing algorithms such as fixed-distance decentralization and a reference anchor point method, the judgment conditions of positive and negative examples are clarified so that the crack image mark data based on the key points of the crack are suitable for model training.

- The image crack detection model KP-CraNet is established. From the perspective of global and local features, the principle of model detection is discussed, and the network structure of the model is introduced. The results show that crack key points greatly improve the crack detection effect.

   A new model evaluation method is proposed. The distance distribution map is used to evaluate the model detection effect based on the key points of the fracture. This research evaluated the model's detection effect through a distance distribution map and the accuracy rate, recall rate, and F1 score. It is shown that the identification model has strong crack identification and robustness.

**Author Contributions:** Conceptualization, D.W. and H.C.; methodology, H.C.; software, H.C.; validation, D.W., H.C. and J.C.; formal analysis, D.W.; investigation, H.C.; resources, J.C.; data curation, J.C.; writing—original draft preparation, H.C.; writing—review and editing, D.W.; visualization, H.C.; supervision, J.C.; project administration, J.C.; funding acquisition, D.W. All authors have read and agreed to the published version of the manuscript.

**Funding:** This research was funded by Natural Science Foundation of Shanghai, grant number 18ZR1414500.

**Data Availability Statement:** Data is available at https://github.com/csga11/craData (accessed on 24 November 2021).

**Conflicts of Interest:** The authors declare no conflict of interest.

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
