# Peer review of "Detection Based on Crack Key Point and Deep Convolutional Neural Network"

_applsci, doi:10.3390/app112311321_

Round 1
Reviewer 1 Report
The manuscript describes an approach for image-based damage detection, resorting to a new concept named ‘crack keypoint’ and anchor points, named “reference anchor point method”. It also proposes the evaluation of distribution map of distance to verify the image crack detection model presented in this study.
Overall, the topic of image-based crack detection and assessment is of great interest to the community of structural engineers. However, before being fully accepted, the paper needs to address several weak points, concerning both its format and content. Specifically:
- The paper is quite lengthy. The authors may consider synthesising some of their sections.
- The abstract as well can be slightly shortened.
- For the same reason as above, almost all figures are also excessively large – specifically Figures 1, 2, 3, 4, 5, 8, 9, 10, 11, 12, 13, 14 can be modified to fit in a much smaller space. Figure 7 may be even omitted for brevity’s sake.
- The concept itself of “crack characterization”, as intended by the Authors in Section 2, is semantically similar to the very classic, well-known Rytter classification of levels of damage identification, which defines the damage (including surface cracks) assessment tasks as follows:
Level 1: Determination that damage is present in the structure
Level 2: Determination of the geometric location of the damage
Level 3: Quantification of the severity of the damage
Level 4: Prediction of the remaining service life of the structure
Since this can confuse the reader, it would be better to explicitly differentiate them in the text. Rytter’s hierarchy was applied to define different video- and image-based Structural Health Monitoring tasks e.g. in https://doi.org/10.1111/str.12336.
- According to Section 3.1, the KP-CraNet KP-CraNet is ResNet with the last three feature layers repurposed for the specific task of crack keypoint feature extraction. This should be (briefly) anticipated in the abstract, where the name “KP-CraNet” is firstly introduced with almost no context.
- In image- and video-based ML approaches for Structural Health Monitoring, differences in illumination, rotation, and angle of the camera point of view can strongly affect the final results. This point should be at least discussed in the manuscript. For instance, Speeded-Up Robust Features (SURF) were tested in https://doi.org/10.1088/1742-6596/1249/1/012004 and used in https://doi.org/10.3390/s19102345 for target tracking in video-based damage detection. The Authors may consider adding these and other similar references.
- Video processing is very computationally expensive. The Authors should investigate and discuss how their approach compare to the state-of-the-art methods reviewed under this aspect (e.g. by comparing the elapsed time).
- The method was trained and tested on a self-produced dataset. However, there is plenty of publicly available datasets to this aim, e.g. https://doi.org/10.17632/5y9wdsg2zt.2. Many other similar databases can be found online for cracks in concrete, road pavements, etc. The Authors may use these available images to allow other researchers to compare their results.
Other minor issues:
- The affiliation number ‘1’ is missing in the author list.
- In many points throughout the whole text, the Authors forgot to add a blank space between the Authors’ surnames and the reference (e.g. O’Byrne[18]). Please correct.
- Page 3 line 128, “Error! Reference source not found.”.
- The equation numbers on the right of each equation are not uniformly spaced for all equations.
- The paper cross-references are not always correct. E.g., on page 18 line 500, it is written “Equation 11” when it clearly refers to Eq 10. There are many other similar cases. Please double-check throughout the whole manuscript
- Even if overall relatively well-written, there are several typos and grammar errors throughout the text. For instance, on page 18 line 469, the definition of \alpha is missing. Please double-check throughout the whole manuscript
- Figures 11 and 12 (all subplots): the x labels have a typo (“Distance”). Moreover, the measurement units should be reported.
- In the Conclusions, it is not clear if the last paragraph (lines 633-637) was intended as the last bullet point, which went lost for some reason.
Reviewer 2 Report
This manuscript reports the development of a crack detection method employing a CNN.
There are plenty of image-based crack detection methods as well as a flourishing field of image detection and analysis methods using CNNs. Therefore, the current work aims to stand out by introducing the concept of crack keypoint, combined with the anchor mechanism.
The abstract is not clearly written. This hinders the readers' perception and negatively affects the judgement of the manuscript.
The theme is current and interesting, even if many good quality solutions are available, even at the commercial level.
Therefore, even if the problem is well defined, the current methods shortcomings should be further discussed to justify a research gap.
The Introduction is not as concise as it should. It ends up being lengthy, yet without being an extensive contribution. The embedded literature review is interesting and has some of the most important references, but many more could have been added and, more importantly, it could have had a much deeper critical analysis.
Moreover, the use of non-scientific language as "potentially bad influence on structural safety" or "through all kinds of sensors" can only be negatively appreciated.
Research methods are adequate and thoroughly explained.
Datasets and results seem to have not been made publicly available. This hiders reproducibility and, without it, the manuscript value is significantly reduced.
Conclusions can be regarded as excessive as, in my opinion, the proposal of a method should be validated (also) against public databases and existing methods.
There are some critical comments to be made about the manuscript readability and intelligibility. Regarding this aspect, the manuscript must be significantly improved. Several redaction issues have been spotted. To name a few, one can mention the curious use of random numbers in the text, in "infrastructure1" or "raised4and", referencing issues as in "source not found", or the weak command of English writing as with the use of "prejudices".
In conclusion, I believe that this is valuable research, but the manuscript needs significant improvements, and data must be made available.
Round 2
Reviewer 1 Report
The authors have now addressed all of the reviewer's main concerns. They also performed all the minor corrections to the text and figures, and tables as suggested.
For what concerns the content of the paper, this Reviewer is satisfied with the corrections made by the Authors. Therefore, the paper can be accepted for publication.
However, there are still some typos and small mistakes throughout the text – e.g., page 2 line 57, “requirements of ap-plying crack detection”; page 3 line 107 “crack feature infor-mation”, and elsewhere. Thus, a further grammar check would be helpful.
Some inline equations (e.g. line 153, page 4) seem to be not perfectly in line with the text as well. However, these issues can be solved in the proofreading process.
Author Response
Dear reviewer,
Thank you very much for your valuable suggestions during the whole process. We have corrected all the problems, and also used the English editing service provided by MDPI to polish the English language style.

Reviewer 2 Report
I thank the Authors for the manuscript enhancement.
As I already said, I do not have significant concerns over the research, but believe that the manuscript should be improved. Now I regard that an effort was made in that direction, and will not push any further, since I believe that the global level is acceptable.
Please make sure that data is made publicly available when the manuscript is published.
Author Response
Dear reviewer,
Thank you very much for your valuable suggestions throughout the process. We have attached the data to the paper (line 411 on page 12), and also used the English editing service provided by MDPI to polish the English language style.
